# Chromoblastomycosis and phaeohyphomycotic abscess-associated hospitalizations, United States, 2016–2021

Dallas J. Smith[1]*, Kaitlin Benedict[1], Shawn R. Lockhart[1], Sanjay G. Revankar[2]

1 Mycotic Diseases Branch, Centers for Disease Control and Prevention, Atlanta, Georgia, United States of America, 2 Office of Infectious Diseases, Center for Drug Evaluation and Research, U.S. Food and Drug Administration, Silver Spring, Maryland, United States of America

* rhq8@cdc.gov

## Abstract

### Background

Chromoblastomycosis and phaeohyphomycotic abscesses are infections of the skin and subcutaneous tissues caused by dematiaceous fungi; more rarely, phaeohyphomycotic brain abscesses can occur. The epidemiology and clinical outcomes of chromoblastomycosis and phaeohyphomycotic abscesses are not well-understood in the United States.

### Methodology/ principal findings

We used data from the Healthcare Cost and Utilization Project's National Inpatient Sample to obtain yearly national estimates of chromoblastomycosis and phaeohyphomycotic abscess-associated hospitalizations. We examined age group, sex, Census region, season of hospital admission, clinical form of chromoblastomycosis, and presence of selected concurrent conditions. An estimated 690 chromoblastomycosis and phaeohyphomycotic abscess-associated hospitalizations occurred during 2016–2021. Rates were highest in 2016 (0.5/1,000,000) and lowest in 2020 (0.2/1,000,000). Overall, higher hospitalization rates occurred among males (0.4/1,000,000) versus females (0.3/1,000,000). Rates increased with age, with patients aged ≥65 years having the highest rate (0.9/1,000,000). The Northeast had the highest hospitalization rate (0.5/1,000,000) followed by the South (0.4/1,000,000). Hypertension (34%), diabetes (33%), dyslipidemia (28%), and chronic kidney disease (21%) were the most common concurrent conditions. Nine percent had autoimmune inflammatory disease or solid malignancy. Seven percent had solid organ or stem cell transplantation. Subsequently, five percent had lymphedema. Mean hospitalization length was 9.9 days; in-hospital death occurred in 3%.

**Data availability statement:** National Inpatient Sample (NIS) data are available from the Healthcare Cost and Utilization Project (HCUP) at https://hcup-us.ahrq.gov/. ICD-10 codes used in this analysis can be round in Supplementary Table 1 (S1 Table).

**Funding:** The author(s) received no specific funding for this work.

**Competing interests:** The authors have declared that no competing interests exist.

## Conclusions/significance

Substantial in-hospital mortality and complications like lymphedema can occur from chromoblastomycosis and phaeohyphomycotic abscesses. Our analysis provides a baseline to monitor hospitalizations and mortality along with comorbidities that may change these outcomes. Public health and clinical partnerships could improve understanding of these fungal diseases caused by dematiaceous fungi through registries, enhanced surveillance, and increased awareness.

## Author summary

Chromoblastomycosis and phaeohyphomycotic abscesses are fungal infections caused by dematiaceous fungi. Usually, these infections cause severe skin and tissue infections but can also cause infections in the brain. Risk factors and hospitalization and mortality rates from these diseases in the United States are not well understood. We used a large administrative database (Healthcare Cost and Utilization Project) to look at hospitalizations caused by chromoblastomycosis and phaeohyphomycotic abscesses. We found 690 chromoblastomycosis and phaeohyphomycotic abscess-associated hospitalizations during 2016–2021. Rates of hospitalizations were higher in males, people aged ≥65 years, and in the Northeast and South regions. People who were hospitalized with these fungal diseases had hypertension (34%), diabetes (33%), dyslipidemia (28%), and chronic kidney disease (21%). Five percent had lymphedema. The average time patients spent hospitalized was 9.9 days, and 3% of patients died. Although rare, given the substantial morbidity and mortality associated with chromoblastomycosis and phaeohyphomycotic abscesses hospitalizations, earlier diagnosis and better treatment is critical. Public health and clinical partnerships could improve understanding of these fungal diseases caused by black fungi.

## Introduction

Chromoblastomycosis and phaeohyphomycotic abscesses (a form of phaeohyphomycosis) are infections caused by dematiaceous fungi that typically affect the skin and subcutaneous tissues. Phaeohyphomycotic abscesses can also affect the brain, leading to high mortality rates [1]. Additionally, phaeohyphomycosis can disseminate to cause endocarditis, fungemia, and pulmonary infections [2]. These fungal diseases are typically acquired after trauma, which allows the fungi to enter into the skin from environmental sources. Phaeohyphomycotic brain abscesses may spread from a primary skin infection or by transient pulmonary acquisition [3]. Chromoblastomycosis is distinguished from phaeohyphomycotic abscesses by microscopically visualizing the characteristic thick-walled, single, or multicellular clusters of pigmented fungal cells (also known as medlar bodies, muriform cells or sclerotic bodies).

Clinical presentation and causative species cannot reliably distinguish between these two fungal diseases and causative species overlap. However, *Exophiala*, *Alternaria*, *Curvularia*, and *Exserohilum* species more commonly cause phaeohyphomycotic abscesses, while *Fonsecaea pedrosoi*, *F. monophora*, *F. nubica*, *Cladophialophora carrionii*, *Rhinocladiella aquaspersa*, and *Phialophora verrucosa* more commonly cause chromoblastomycosis [4,5]. Phaeohyphomycotic brain abscesses are most often caused by *Cladophialophora bantiana,* typically in immunocompetent hosts [1,2]. Chromoblastomycosis may occur more commonly in immunocompetent hosts and phaeohyphomycotic abscesses in immunocompromised hosts [2,6]. Chromoblastomycosis is recognized as a neglected tropical disease by the World Health Organization [7].

The epidemiology and clinical outcomes of chromoblastomycosis and phaeohyphomycotic abscesses are not well-understood in the United States. An analysis using insurance claims data found a 5-year prevalence rate of chromoblastomycosis and phaeohyphomycotic abscesses of 14.7/1,000,000 patients, however, this was based on people with commercial insurance and clinical outcomes could not be assessed [8]. An active surveillance program in the San Francisco area estimated a yearly incidence of 1/1,000,000 patients for phaeohyphomycosis [9]. A prospective study of 99 phaeohyphomycosis cases in the United States, Peru, and Australia found an all-cause mortality rate of 16% at 30 days [2].

To understand trends and comorbidities for chromoblastomycosis and phaeohyphomycotic abscess-associated hospitalizations, we analyzed 2016–2021 data from the Healthcare Cost and Utilization Project (HCUP), a set of databases sponsored by the Agency for Healthcare Research and Quality.

## Methods

The HCUP National Inpatient Sample (NIS) is the largest publicly available all-payer database of hospital inpatient stays in the United States. The NIS represents a 20% stratified sample of all discharges from U.S. community hospitals, excluding rehabilitation and long-term acute care hospitals. To enable calculation of national estimates, discharge-level weights are assigned based on the hospital's U.S. Census division, ownership, urban/rural location, teaching status, and number of beds. The unweighted NIS contains data on approximately 7 million inpatient stays each year, which translates to approximately 35 million inpatient stays when weighted.

We identified chromoblastomycosis and phaeohyphomycotic abscess-associated hospitalizations and concurrent conditions using selected ICD-10-CM codes listed anywhere on the discharge record (S1 Table).

We obtained yearly national estimates of chromoblastomycosis and phaeohyphomycotic abscess-associated hospitalizations using the HCUP-supplied discharge weights and examined by age group, sex, Census region [10]), season of hospital admission, clinical form of chromoblastomycosis (ICD-10-CM codes B43.0, B43.8, and B43.9) and phaeohyphomycotic abscess (ICD-10-CM codes B43.1 and B43.2), and presence of selected concurrent conditions using SAS 9.4 (SAS Institute, Cary, NC, USA) survey procedures (PROC SURVEYFREQ and PROC SURVEYMEANS). Disseminated phaeohyphomycosis, other than brain abscesses, could not be captured in our analysis as it does not have a specific ICD-10-CM code and is often captured with another non-specific code (B48.8, other specified mycoses). Overall rates were calculated using population estimates from the U.S. Census Bureau during 2016–2021.

### Ethical statement

This activity was reviewed by CDC and was conducted consistent with applicable federal law and CDC policy (See, e.g.,: 45 C.F.R. part 46, 21 C.F.R. part 56; 42 U.S.C. §241(d); 5 U.S.C. §552a; 44 U.S.C. §3501 et seq).

## Results

Six-hundred and ninety chromoblastomycosis and phaeohyphomycotic abscess-associated hospitalizations occurred during 2016–2021. Most hospitalizations were for subcutaneous phaeohyphomycotic abscess and cyst (n = 395, 57%) or

**Table 1. Number of chromoblastomycosis and phaeohyphomycotic abscess-associated hospitalizations by ICD-10-CM B43 subtype code, 2016 to 2021.**

| Characteristic | Total | |
|---|---|---|
| | n | Column percentage |
| Total | 690 | 100% |
| **ICD-10-CM B43 subtype code** | | |
| Cutaneous chromoblastomycosis (B43.0) | 155 | 22% |
| Phaeohyphomycotic brain abscess (B43.1) | 40 | 6% |
| Subcutaneous phaeohyphomycotic abscess and cyst (B43.2) | 395 | 57% |
| Other forms of chromoblastomycosis (B43.8) | 30 | 4% |
| Chromoblastomycosis, unspecified (B43.9) | 70 | 10% |

cutaneous chromoblastomycosis (n = 155, 22%) (Table 1). Rates were highest in 2016 (0.5/1,000,000) and lowest in 2020 (0.2/1,000,000) (Table 2).

Overall, higher hospitalization rates occurred among males (0.4/1,000,000) versus females (0.3/1,000,000). Rates increased with age, with patients aged ≥65 years having the highest rate (0.9/1,000,000). The Northeast had the highest hospitalization rate (0.5/1,000,000) followed by the South (0.4/1,000,000). By season of hospital admission, most hospitalizations for chromoblastomycosis alone occurred in Fall (37%). Summer (28%) had the largest proportion of admissions for phaeohyphomycotic abscesses.

Chromoblastomycosis and phaeohyphomycotic abscess was listed as the primary discharge diagnosis in 6% of chromoblastomycosis and phaeohyphomycotic abscess-associated hospitalizations. Hypertension (34%), diabetes (33%), dyslipidemia (28%), and chronic kidney disease (21%) were the most common selected concurrent conditions. Nine percent each had autoimmune inflammatory disease or solid malignancy. Seven percent had solid organ or stem cell transplantation. Lymphedema was identified in 14% of chromoblastomycosis hospitalizations; no patients with phaeohyphomycotic abscess-related hospitalizations had lymphedema.

Mean hospitalization length was 9.9 days (standard error [SE] 1.5 days); 10.7 days (SE 2.1) for phaeohyphomycotic abscesses and 8.5 days (SE 1.8) for chromoblastomycosis. In-hospital death occurred in 3% of all chromoblastomycosis and phaeohyphomycotic abscess-associated hospitalizations.

## Discussion

Hospitalizations from chromoblastomycosis and phaeohyphomycotic abscesses are rare in the United States but can lead to complications like lymphedema. Substantial in-hospital mortality can occur from these fungal diseases (largely phaeohyphomycotic abscesses) with 3% seen our study; another study saw all-cause mortality of 16%, although, this included disseminated phaeohyphomycosis [2]. Our analysis provides a baseline to monitor hospitalizations and mortality along with changes in risk factors that may impact these outcomes. These limited data support the development of prospective registries that can inform treatment recommendations and better patient management to decrease mortality rates.

Hospitalization rates were low and fairly consistent across the study period. As these diseases seldom require immediate care, the COVID-19 pandemic may have changed healthcare seeking behavior, particularly access to dermatologists, for subcutaneous phaeohyphomycotic abscesses and cysts and cutaneous chromoblastomycosis and led to the lowest rate in 2020 [11]. Strong seasonal patterns of hospital admissions were not evident at a national level from our analysis. The higher proportion of hospital admissions in the Fall (chromoblastomycosis) and Summer (phaeohyphomycotic abscesses) may be related to increased outdoor activities during these warmer months which could offer more opportunities for trauma and implantation of the fungus from the environment. However, more likely, these slight seasonal trends were obscured from the indolent nature of these diseases and associated delayed diagnoses and long incubation periods.

**PLOS Neglected Tropical Diseases**

**Table 2. Hospitalization rates among patients with chromoblastomycosis and phaeohyphomycotic abscess, 2016 to 2021.**

| Characteristic | Total | | | Chromoblastomycosis | | Phaeohyphomycotic abscesses | |
|---|---|---|---|---|---|---|---|
| | N | Rate/1,000,000 | col % | n | col % | n | col % |
| **Total** | 690 | 0.4 | 100% | 255 | 100% | 435 | 100% |
| **Year** | | | | | | | |
| 2016 | 175 | 0.5 | 25% | 40 | 16% | 135 | 31% |
| 2017 | 119 | 0.4 | 17% | 50 | 20% | 70 | 16% |
| 2018 | 130 | 0.4 | 19% | 55 | 22% | 75 | 17% |
| 2019 | 85 | 0.3 | 12% | 45 | 18% | 40 | 9% |
| 2020 | 69 | 0.2 | 10% | 30 | 12% | 40 | 9% |
| 2021 | 110 | 0.3 | 16% | 35 | 14% | 75 | 17% |
| **Mean age, years (std err)** | 56.4 | | (1.8) | 58.4 | (2.4) | 55.2 | (2.1) |
| **Age group, years** | | | | | | | |
| <18 | 50 | 0.1 | 7% | 15 | 6% | 35 | 8% |
| 18–44 | 100 | 0.1 | 14% | 35 | 14% | 65 | 15% |
| 45–64 | 255 | 0.5 | 37% | 90 | 35% | 165 | 38% |
| ≥65 | 285 | 0.9 | 41% | 115 | 45% | 170 | 39% |
| **Sex** | | | | | | | |
| Male | 405 | 0.4 | 59% | 155 | 61% | 250 | 57% |
| Female | 285 | 0.3 | 41% | 100 | 39% | 185 | 43% |
| **Hospital region** | | | | | | | |
| Northeast | 155 | 0.5 | 22% | 55 | 22% | 100 | 23% |
| Midwest | 125 | 0.3 | 18% | 55 | 22% | 70 | 16% |
| South | 320 | 0.4 | 46% | 125 | 49% | 195 | 45% |
| West | 90 | 0.2 | 13% | 20 | 8% | 70 | 16% |
| **Race/ethnicity (n=655)** | | | | | | | |
| White | 440 | | 64% | 170 | 67% | 270 | 62% |
| Black | 130 | | 19% | 35 | 14% | 95 | 22% |
| Hispanic | 50 | | 7% | 15 | 6% | 35 | 8% |
| Other race/ethnicity | 35 | | 5% | 20 | 8% | 15 | 3% |
| **Payer** | | | | | | | |
| Medicare | 350 | | 51% | 140 | 55% | 210 | 48% |
| Medicaid | 110 | | 16% | 35 | 14% | 75 | 17% |
| Private | 215 | | 31% | 75 | 29% | 140 | 32% |
| Other | 15 | | 2% | * | | * | |
| **Concurrent diagnoses** | | | | | | | |
| Alcohol-related disorders | 55 | | 8% | 20 | 8% | 35 | 8% |
| Autoimmune inflammatory disease | 60 | | 9% | 30 | 12% | 30 | 7% |
| COPD | 105 | | 15% | 35 | 14% | 70 | 16% |
| Diabetes | 225 | | 33% | 120 | 47% | 105 | 24% |
| Hematologic malignancy | 25 | | 4% | | | | |
| HIV infection | 0 | | 0% | 0 | 0% | 0 | 0% |
| Lymphedema | 35 | | 5% | 35 | 14% | 0 | 0% |
| Solid malignancy | 60 | | 9% | 30 | 12% | 30 | 7% |
| Solid organ or stem cell transplant | 50 | | 7% | * | | * | |
| Chronic kidney disease | 145 | | 21% | 65 | 25% | 80 | 18% |
| Dyslipidemia | 195 | | 28% | 85 | 33% | 110 | 25% |

*(Continued)*

**Table 2.** (Continued)

| Characteristic | Total N | Rate/1,000,000 | col % | Chromoblastomycosis n | col % | Phaeohyphomycotic abscesses n | col % |
|---|---|---|---|---|---|---|---|
| Hypertension | 235 | | 34% | 125 | 49% | 110 | 25% |
| Liver disease | 50 | | 7% | 30 | 12% | 20 | 5% |
| **ICD-10 B43 was primary diagnosis** | 40 | | 6% | * | | * | |
| **Season** | | | | | | | |
| Winter | 150 | | 22% | 45 | 18% | 105 | 24% |
| Spring | 170 | | 25% | 65 | 25% | 105 | 24% |
| Summer | 170 | | 25% | 50 | 20% | 120 | 28% |
| Fall | 200 | | 29% | 95 | 37% | 105 | 24% |
| **In-hospital death** | 20 | | 3% | * | | * | |
| **Mean length of stay, days (std err)** | 9.9 | | (1.5) | 8.5 | (1.8) | 10.7 | (2.1) |

*Cells with n ≤ 10 hospitalizations are suppressed.

Abbreviations: col %, column percentage; std err, standard error; COPD, chronic obstructive pulmonary disease; HIV, human immunodeficiency virus; ICD, International Classification of Diseases.

Activities related to exposure can be studied further through enhanced surveillance activities to correlate the seasonal findings in our study. Regardless, infections can be acquired throughout the year, and healthcare providers should maintain suspicion for chromoblastomycosis and phaeohyphomycotic abscesses to prevent delayed diagnoses and improve patient outcomes.

Our findings of higher rates of hospitalizations among men and those aged ≥65 years are consistent with the demographic characteristics of non-hospitalized persons with this disease in the United States [8]. For most fungal diseases, men seem to be at greater risk for invasive fungal diseases; hypotheses for these higher rates include behaviors leading to environmental exposure or interactions of sex hormones. [12,13]. The same reasons might explain the higher hospitalization rates with implantation mycoses like chromoblastomycosis and phaeohyphomycotic abscesses among men seen in our study. Higher hospitalization rates in older persons may be related to host immune factors or behavioral factors like increased outdoor activities [14].

Chromoblastomycosis and phaeohyphomycotic abscesses can have long-term complications, including lymphedema, secondary bacterial infections, squamous cell carcinoma, and tissue fibrosis [15,16]. In our study, one of every seven patients diagnosed with chromoblastomycosis experienced lymphedema, which can present chronically, reduce mobility, and create social stigma [6]. Other concurrent conditions seen in our study -- diabetes, solid malignancy, and solid organ transplants -- were also reported as common risk factors in other studies for these diseases [2,8]

Our study has at least three limitations. First, miscoding of these rare diseases is possible. Distinguishing between chromoblastomycosis and phaeohyphomycotic abscesses is difficult and could have led to misclassification. However, this would not affect the overall rate of having one of these two diseases. Second, the NIS does not include outpatient visits, which likely led to over-representation of phaeohyphomycotic abscesses as patients with chromoblastomycosis are less likely to be hospitalized. Third, outcomes related to chromoblastomycosis and phaeohyphomycotic abscesses were not captured fully, as complications such as amputation or tissue fibrosis may have occurred after hospital discharge. The NIS does not enable analyses of repeat hospitalizations among the same patient.

The substantial morbidity and mortality associated with chromoblastomycosis and phaeohyphomycotic abscess-associated hospitalizations renders reducing diagnostic delays and optimizing antifungal therapy critical for patients with these rare conditions. Public health and clinical partnerships could improve understanding of these fungal diseases caused by dematiaceous fungi through registries, enhanced surveillance, and awareness raising.

## Supporting information

**S1 Table. ICD-10 codes for chromoblastomycosis and phaeohyphomycotic abscess and other conditions.**
(DOCX)

## Acknowledgments

We thank the partner organizations who provide data to the Healthcare Cost and Utilization Project (https://www.hcup-us.ahrq.gov/db/hcupdatapartners.jsp). The findings and conclusions in this report are those of the authors and do not necessarily represent the official position of the U.S Centers for Disease Control and Prevention. This article reflects the views of the authors and should not be construed to represent FDA's views or policies.

## Author contributions

**Conceptualization:** Dallas J. Smith, Kaitlin Benedict.

**Data curation:** Kaitlin Benedict.

**Formal analysis:** Kaitlin Benedict.

**Supervision:** Shawn R. Lockhart, Sanjay G. Revankar.

**Writing – original draft:** Dallas J. Smith.

**Writing – review & editing:** Dallas J. Smith, Kaitlin Benedict, Shawn R. Lockhart, Sanjay G. Revankar.

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
