## [Decision Letter · Decision Letter 0]

23 Jun 2025

PNTD-D-25-00604

Chromoblastomycosis and Phaeohyphomycotic Abscess-Associated Hospitalizations, United States, 2016–2021

Dear Dr. Smith,

Thank you for submitting your manuscript to PLOS Neglected Tropical Diseases. After careful consideration, we feel that it has merit but does not fully meet PLOS Neglected Tropical Diseases's publication criteria as it currently stands. Therefore, we invite you to submit a revised version of the manuscript that addresses the points raised during the review process.

Please submit your revised manuscript within 60 days Aug 22 2025 11:59PM. If you will need more time than this to complete your revisions, please reply to this message or contact the journal office at plosntds@plos.org. Please include the following items when submitting your revised manuscript:

We look forward to receiving your revised manuscript.

Kind regards,

Felix Bongomin, MB ChB, MSc, MMed, FECMM

Academic Editor

Marcio Rodrigues

Section Editor

Shaden Kamhawi

co-Editor-in-Chief

Paul Brindley

co-Editor-in-Chief

**Journal Requirements:**

At this stage, the following Authors/Authors require contributions: Dallas J. Smith, Kaitlin Benedict, Shawn R. Lockhart, and Sanjay G. Revankar. Please ensure that the full contributions of each author are acknowledged in the "Add/Edit/Remove Authors" section of our submission form.

2) We notice that your supplementary Table is included in the manuscript file. Please remove it and upload it with the file type 'Supporting Information'. Please ensure that each Supporting Information file has a legend listed in the manuscript after the references list.

3) Please provide a completed 'Competing Interests' statement, including any COIs declared by your co-authors. If you have no competing interests to declare, please state "The authors have declared that no competing interests exist". Otherwise please declare all competing interests beginning with the statement "I have read the journal's policy and the authors of this manuscript have the following competing interests:"

**Comments to the Authors:**

**Please note that two reviews are uploaded as attachments.**

**Reviewers' Comments:**

Reviewer's Responses to Questions

**Key Review Criteria Required for Acceptance?**

**Methods**

-Are the objectives of the study clearly articulated with a clear testable hypothesis stated?

-Is the study design appropriate to address the stated objectives?

-Is the population clearly described and appropriate for the hypothesis being tested?

-Is the sample size sufficient to ensure adequate power to address the hypothesis being tested?

-Were correct statistical analysis used to support conclusions?

-Are there concerns about ethical or regulatory requirements being met?

Reviewer #1: (No Response)

Reviewer #2: Please see attached comments

Reviewer #3: The methodological revisions have been detailed in the attached document

**Results**

-Does the analysis presented match the analysis plan?

-Are the results clearly and completely presented?

-Are the figures (Tables, Images) of sufficient quality for clarity?

Reviewer #1: (No Response)

Reviewer #2: Please see attached comments

Reviewer #3: The data are clearly and concisely presented in tables.

**Conclusions**

-Are the conclusions supported by the data presented?

-Are the limitations of analysis clearly described?

-Do the authors discuss how these data can be helpful to advance our understanding of the topic under study?

-Is public health relevance addressed?

Reviewer #1: (No Response)

Reviewer #2: Please see attached comments

Reviewer #3: The conclusions are relevant, as long as the authors adjust them to reflect a broader scope, appropriately framed as 'melanized fungal infections

**Editorial and Data Presentation Modifications?**

Reviewer #1: (No Response)

Reviewer #2: Please see attached comments

Reviewer #3: Based on the relevance of the topic, the clarity of the data presentation, and the pertinence of the conclusions—pending the suggested adjustments—I recommend the manuscript be accepted with minor revisions.

**Summary and General Comments**

Reviewer #1: Dear Editors and Authors,

Thank you for this opportunity to review this report. Smith and colleagues analyze U.S. hospitalizations related to chromoblastomycosis and phaeohyphomycotic abscesses—rare fungal infections caused by dematiaceous fungi—using national inpatient data from 2016 to 2021. An estimated 690 hospitalizations occurred over the six-year period, with rates highest in 2016 and lowest in 2020. Hospitalization rates were higher among males, older adults (≥65 years), and in the Northeast and South regions. Common comorbidities included hypertension, diabetes, dyslipidemia, and chronic kidney disease. Severe outcomes included lymphedema (5%) and in-hospital death (3%), with an average hospital stay of nearly 10 days.

The aim of this study was to enhance understanding of the epidemiology and clinical outcomes of these rare fungal infections, and thereby highlight the need for increased awareness. I do believe that the authors have been able to accomplish this aim to a certain extent. This study provides important baseline data that can inform future surveillance and management efforts. Some notable strengths include novelty, relevance, and use of a large national dataset increasing the generalizability of findings, coupled with balanced discussion.

While this manuscript is generally well-written, with clear articulation of objectives, methods, and limitations, there are areas in this paper where clarification, improved organization, and enhanced precision could strengthen the work. Please see comments below for specific concerns.

Author lay Summary

When you search black fungi online, results show mucormycosis, as opposed to chromoblastomycosis and phaeohyphomycosis, which is what this research is about. Consider changing black fungi to another term, such as melanized fungi or pigmented fungi.

Introduction

The background effectively introduces the clinical relevance of chromoblastomycosis and phaeohyphomycotic abscesses, noting their rarity and the knowledge gap in U.S. epidemiology. There have been multiple mentions of phaeohyphomycosis affecting the brain but disseminated phaeohypomycosis is also relatively common. It would be helpful to briefly mention the full clinical spectrum of disease by discussing other organ systems besides skin, subcutaneous tissue and brain it involves. This will help readers to contextualize the importance of studying these infections.

Methods

• Second paragraph – at the end of the sentence, include the study period *for example, from January 2016 to December 2021. Also, consider explaining here why disseminated phaeohyphomycoses was not a part of this study, instead of outlining coding limitations at the very end.

• The link for Census region can be moved to references.

• If data is available, it would be helpful to add (1) the antifungal/s used and (2) % of patients undergoing surgical resection. Presenting mortality, length of stay, or complications stratified by diagnosis group (chromoblastomycosis vs. phaeohyphomycotic abscess) could add more clinical value, even if limited by small sample sizes as mortality is generally substantially lower in chromoblastomycosis.

• Please address typographical redundancy – “B43.8 and B43.8.”

• It would be helpful to add the reference to the SAS 9.4 survey procedure.

Results

• “Rates were highest in 2016 (0.5/1,000,000) and lowest in 2020 (0.2/1,000,000). Overall, higher hospitalization rates occurred among males (0.4/1,000,000) versus females (0.3/1,000,000) (Table 2). Rates increased with age, with patients aged ≥65 years having the highest rate (0.9/1,000,000). The Northeast had the highest hospitalization rate (0.5/1,000,000) followed by the South (0.4/1,000,000).” - consider rephrasing these sentences as they are the same as abstract. Also, would add (Table 2) after the first sentence instead of the second sentence.

• “By season admitted, more patients were admitted in Fall (29%) which was higher (37%) for chromoblastomycosis alone. Summer (28%) had the largest proportion of admissions for phaeohyphomycotic abscesses.” – consider rephrasing this sentence as it is unclear and can be easily misinterpreted.

Discussion

• Change Discussions to Discussion

• Lymphedema is listed as comorbidities in the abstract, results and table. However, in the discussion it is listed as a complication. Would urge the authors to be consistent. For the table, consider moving lymphedema out from comorbidity and add another column as complication and list lymphedema there.

• Please explain what you mean by – “Our analysis provides a baseline to monitor hospitalizations and mortality along with changes in comorbidities that may impact these outcomes.” What changes in comorbidities?

• “Chromoblastomycosis and phaeohyphomycotic abscesses can have long-term complications, including lymphedema, secondary bacterial infections, squamous cell carcinoma, and tissue fibrosis (14,15).” – Would be helpful to evaluate what % of patients in your study developed secondary bacterial infections, as I understand amputation, SCC, and tissue fibrosis can occur after discharge as outlined in your limitation.

• “Other concurrent conditions seen in our study -- diabetes, solid malignancy, and solid organ transplants -- were also reported as common risk factors in other studies on chromoblastomycosis and phaeohyphomycotic abscesses (6,8)” – would move this section above complications and separate comorbidities from complications.

• The phrase “chromoblastomycosis and phaeohyphomycotic abscess-associated hospitalizations” is repeated excessively; consider using abbreviations or alternate phrasing for brevity once the terms are defined.

Tables

Table 1

• What was the difference between other forms of chromomycosis and Chromomycosis, unspecified? Can consider explaining in methods.

• The total of individual percentages is not adding to 100%

• At the end of the title of table 1, would say – by B43 subtype “of ICD-10-CM code”, 2016 to 2021; as it’s not clear what B43 subtype entails if you simply read the title of the table

• Please spell out coI%. If it is confidence interval, include 95% confidence intervals for key rates if possible.

Table 2

• The total of individual percentages is not adding to 100% for several cells

• Please spell out col%. If it is confidence interval, include 95% confidence intervals for key rates if possible.

Inclusion of at least one epidemiological figure or graph (e.g., annual trends, geographic variation) would improve accessibility and visual engagement.

Reviewer #2: Please see attached comments

Reviewer #3: (No Response)

PLOS authors have the option to publish the peer review history of their article (what does this mean? ). If published, this will include your full peer review and any attached files.

**Do you want your identity to be public for this peer review?** For information about this choice, including consent withdrawal, please see our Privacy Policy .

Reviewer #1: **Yes: ** Tulip A. Jhaveri, MD, FACP, FIDSA

Reviewer #2: **Yes: ** Flavio Queiroz-Telles

Reviewer #3: **Yes: ** DANIEL WAGNER DE CASTRO LIMA SANTOS

**Figure resubmission:**
---

## [Decision Letter · Decision Letter 1]

23 Aug 2025

Dear Dr. Smith,

We are pleased to inform you that your manuscript 'Chromoblastomycosis and Phaeohyphomycotic Abscess-Associated Hospitalizations, United States, 2016–2021' has been provisionally accepted for publication in PLOS Neglected Tropical Diseases.

Best regards,

Felix Bongomin, MB ChB, MSc, MMed, FECMM

Academic Editor

Marcio Rodrigues

Section Editor

Shaden Kamhawi

co-Editor-in-Chief

Paul Brindley

co-Editor-in-Chief

Reviewer's Responses to Questions

**Key Review Criteria Required for Acceptance?**

**Methods**

-Are the objectives of the study clearly articulated with a clear testable hypothesis stated?

-Is the study design appropriate to address the stated objectives?

-Is the population clearly described and appropriate for the hypothesis being tested?

-Is the sample size sufficient to ensure adequate power to address the hypothesis being tested?

-Were correct statistical analysis used to support conclusions?

-Are there concerns about ethical or regulatory requirements being met?

Reviewer #1: (No Response)

Reviewer #2: This manuscript cab be accept for publication

**Results**

-Does the analysis presented match the analysis plan?

-Are the results clearly and completely presented?

-Are the figures (Tables, Images) of sufficient quality for clarity?

Reviewer #1: (No Response)

Reviewer #2: Yes

**Conclusions**

-Are the conclusions supported by the data presented?

-Are the limitations of analysis clearly described?

-Do the authors discuss how these data can be helpful to advance our understanding of the topic under study?

-Is public health relevance addressed?

Reviewer #1: (No Response)

Reviewer #2: Yes

**Editorial and Data Presentation Modifications?**

Reviewer #1: (No Response)

Reviewer #2: In my opinion the authors incorporated most of the reviewers suggestions or corrections

**Summary and General Comments**

Reviewer #1: The authors have addressed all comments on my end, and I believe this manuscript is suitable for publication.

Reviewer #2: I have no comments to add

PLOS authors have the option to publish the peer review history of their article (what does this mean? ). If published, this will include your full peer review and any attached files.

**Do you want your identity to be public for this peer review?** For information about this choice, including consent withdrawal, please see our Privacy Policy .

Reviewer #1: **Yes: ** Tulip A. Jhaveri

Reviewer #2: **Yes: ** Flavio Queiroz-Teles

---

## [Editor Report · Acceptance letter]

Dear Dr. Smith,

We are delighted to inform you that your manuscript, " 

Chromoblastomycosis and Phaeohyphomycotic Abscess-Associated Hospitalizations, United States, 2016–2021," has been formally accepted for publication in PLOS Neglected Tropical Diseases.

Best regards,

Shaden Kamhawi

co-Editor-in-Chief

Paul Brindley

co-Editor-in-Chief
